# Strategies for Successful Over-Expression of Human Membrane Transport Systems Using Bacterial Hosts: Future Perspectives

**DOI:** 10.3390/ijms23073823

**Published:** 2022-03-30

**Authors:** Michele Galluccio, Lara Console, Lorena Pochini, Mariafrancesca Scalise, Nicola Giangregorio, Cesare Indiveri

**Affiliations:** 1Unit of Biochemistry and Molecular Biotechnology, Department of Biology, Ecology and Earth Sciences (DiBEST), University of Calabria, Via P. Bucci 4c, Arcavacata di Rende, 87036 Cosenza, Italy; michele.galluccio@unical.it (M.G.); lara.console@unical.it (L.C.); lorena.pochini@unical.it (L.P.); mariafrancesca.scalise@unical.it (M.S.); 2Institute of Biomembranes, Bioenergetics and Molecular Biotechnology (IBIOM), National Research Council (CNR), Via Amendola 165/A, 70126 Bari, Italy; n.giangregorio@ibiom.cnr.it

**Keywords:** SLC transporters, over-expression, *E. coli*, channels, ABC transporters

## Abstract

Ten percent of human genes encode for membrane transport systems, which are key components in maintaining cell homeostasis. They are involved in the transport of nutrients, catabolites, vitamins, and ions, allowing the absorption and distribution of these compounds to the various body regions. In addition, roughly 60% of FDA-approved drugs interact with membrane proteins, among which are transporters, often responsible for pharmacokinetics and side effects. Defects of membrane transport systems can cause diseases; however, knowledge of the structure/function relationships of transporters is still limited. Among the expression of hosts that produce human membrane transport systems, *E. coli* is one of the most favorable for its low cultivation costs, fast growth, handiness, and extensive knowledge of its genetics and molecular mechanisms. However, the expression in *E. coli* of human membrane proteins is often toxic due to the hydrophobicity of these proteins and the diversity in structure with respect to their bacterial counterparts. Moreover, differences in codon usage between humans and bacteria hamper translation. This review summarizes the many strategies exploited to achieve the expression of human transport systems in bacteria, providing a guide to help people who want to deal with this topic.

## 1. Introduction

Membrane proteins are crucial for life, because they allow the various body regions to communicate with each other. The importance of this particular class of proteins is also demonstrated by the fact that 20 to 30% of the ORFs of sequenced genomes encode membrane proteins [1,2]; this value is 26% for the human genome [3]. Among membrane proteins, transporters, which account for 10% of the entire human genome [4], play the specific role of mediating the flux of nutrients such as sugars, amino acids, lipids, ions, vitamins, and catabolites through plasma and sub-cellular membranes. Derangements or malfunctioning of membrane transport systems can cause a wide range of diseases [5]. In particular, the entries retrieved from the Online Mendelian Inheritance in Man (OMIM) database using the generic keywords of “channel” or “transporter” amount to more than two thousand, highlighting the importance of studying these proteins [6]. Thus, the identification of membrane transporters with aberrant properties can result in the discovery of novel therapeutic targets [7,8]. In addition to their physiological roles, many transporters are involved in drug interactions, absorption, and excretion. Indeed, about 60% of FDA-approved drug targets are membrane proteins, including a growing number of transporters [9,10]. In this scenario, knowledge of transporter functions and structure/function relationships are fundamental not only for basic knowledge, but also for application to human health. Despite the importance of membrane transport proteins, the knowledge of members of this wide group in humans is still limited due to the difficulty in expressing, purifying, and assaying the function of these kinds of proteins [11]. To obtain a human transport proteins suitable for functional and structural studies, several expression hosts, either prokaryotic or eukaryotic, have been adopted; among these are, *E. coli* [12], *L. lactis* [13], *S. cerevisiae* [14], *P. pastoris* [15], *S. frugiperda* [16], and HEK293F [17]. While several membrane proteins have been more successfully expressed using non-bacterial hosts, albeit with low yields, bacterial over-expression is a powerful tool for studying membrane transporters, since it allows researchers to obtain purified human proteins at a low cost. However, *E. coli* frequently shows poor tolerance to the expression of human membrane proteins. Indeed, these proteins have intrinsic hydrophobicity, which makes them prone to aggregating. *E. coli* is not equipped with chaperone systems that recognize human hydrophobic domains, thus strengthening its propensity to aggregate. The expressed human transport proteins often form inclusion bodies (IBs) from which it is difficult to recover the protein in an active folded form [18,19]. Thus, many strategies have been adopted to overcome the described difficulties. To prevent IB formation, a possible solution is to add a specific tag improving the solubility of the expressed protein [20] or a periplasmic signal sequence for membrane or periplasmic targeting, respectively [21,22]. However, in cases of toxic targets, IB formation can be desirable since they are generally mechanically stable and protected against proteolytic degradation, and can be isolated from bacterial cells by sucrose-gradient centrifugation [23]. In these cases, IB formation strongly boosts the production of a membrane protein up to several-thousand-fold without toxicity for bacteria [24,25]. *E. coli* is unable to perform a large number of the post-transcriptional modifications that normally occur in human proteins, such as glycosylation. To solve this problem, *Campylobacter jejuni* N-glycosylation machinery was transformed in *E. coli* for glycoprotein production [26,27]. Moreover, another important issue to overcome is the different codon usage of *E. coli* with respect to that of eukaryotic organisms, i.e., a different frequency of the occurrence of synonymous codons [28]. This is commonly solved using *E. coli* strains with a codon supply or by optimizing cDNA codon usage [28,29,30]. Altogether, the optimization of every single step of the protocols for the production of human membrane transporters—such as strain selection, vector design (specific tag and its relative position), cultivation conditions, and expression induction—often have a significant and positive impact on the final yield and sample quality (Figure 1) [11].

This review summarizes the solutions that have been adopted so far to obtaining successful bacterial over-expression of human transport systems, with the aim of providing insight for undertaking the expression of a novel membrane transport system.

## 2. Membrane Transport Systems: A Brief Overview

As stated in Section 1, membrane transport systems are a key element in cell life, since they regulate the flux of nutrients, metabolites, and toxic compounds through cell membranes. Based on the type of transported molecules and on the energy coupling for driving transport down or against concentration gradients, transport systems are divided into three groups: (i) solute carriers (SLCs); (ii) ATP-dependent transporters; and (iii) ion channels [4,31].

### 2.1. Solute Carrier (SLC)

The SLC superfamily currently includes more than 450 transport proteins grouped into 65 families that transport a wide variety of substances across cell membranes [31,32]. Within each family, protein members share at least 20–25% sequence similarity with at least one other member of the family [32]. SLCs transport a wide variety of molecules, including sugars, amino acids, vitamins, nucleotides, inorganic and organic ions, oligopeptides, and xenobiotics/drugs [4]. Among the SLC transporter superfamily, some members exhibit broad substrate specificity, and others transport only one or a restricted group of biomolecules. Many members are still ‘orphans’, with no known substrate. The SLC superfamily includes either (i) facilitative transporters that act as a simple gatekeeper for a compound that passively moves down its concentration gradient, or (ii) secondary active transporters moving two substrates; one substrate moves down its electro/chemical gradient, providing the free energy to drive the transport of the other substrate against the concentration gradient. A hydropathy plot analysis performed on SLC proteins showed that they contain one to sixteen transmembrane (TM) domains, although most (~83%) are characterized by seven to twelve TM domains [33]. Two of the most common structural folds for SLC members are the major facilitator superfamily (MFS) and the leucine transporter (LeuT)-like folds [33]. Due to the difficulty in expressing and purifying these proteins, relatively few high-resolution structures for human SLCs have been solved [31]. To date, only 41 out of 458 (8.95%) are available in the PDB database. To unravel new molecular and functional aspects of known SLC transporters and to deorphanize the still unknown members, it is crucial to over-express the protein for performing in vitro functional assays. In the following sections, the strategy used for the bacterial expression of SLC transporters is described.

### 2.2. ABC Transporters

The ABC is one of the biggest protein transporter superfamilies present in all living organisms [34,35]. Human ABC transporters possess an ATP-binding cassette, also known as the ‘nucleotide-binding domain’ (NBD). The NBD presents several highly conserved motifs, including the Walker A and Walker B sequences, the ABC signature motif, the H loop, and the Q loop. Besides the NDB domain, ABC transporters also contain trans-membrane domains (TMDs) characterized by several hydrophobic α-helices. The ABC transporter core unit consists of two NBDs that bind and hydrolyze ATP, and two TMDs that are involved in substrate recognition and translocation across the lipid membrane [36]. In humans, there are 49 known ABC genes classified into seven different families (A–G) depending on their amino acid sequence and, eventually, on their protein domains [37]. Mutations in at least 11 of these genes are already known to cause severe inherited diseases such as cystic fibrosis and X-linked adrenoleukodystrophy (X-ALD) [36]. A variety of substances such as ions, carbohydrates, lipids, xenobiotics, and drugs are actively transported out of cells or into cellular organelles [38]. Most human ABC transporters are expressed as full-length proteins, whereas seven members of the B family, four of the D family, and five of the G family are expressed as “half-transporters” that must dimerize to be functional [36,37]. In the following sections, we focus on human ABC transporter family members that are expressed in *E. coli* and functionally characterized.

### 2.3. Channels

Ion channels are integral membrane proteins that selectively conduct ions across the cellular membranes. Ion channels possess common structural characteristics that include a water-filled permeation pathway, also called a pore, which allows ions to flow across the cell membrane, and a selectivity filter, which is the narrowest part of the pore and is responsible for ion species selection [39]. Ion channels have a gating mechanism regulating the switch between the open and closed conformations. On the basis of the switch regulator, ion channels can be divided into voltage-gated ion channels, which are regulated by changes in membrane potential, and ligand-gated ion channels, modulated by the binding of a ligand such as a hormone or a neurotransmitter. Ion channels are ubiquitous and are involved in the regulation of many physiological events such as excitability, contraction, cell-cycle progression, and metabolism. Thus, defects in ion channel function can impair important biological processes, leading to a wide range of diseases [40].

## 3. Optimization of Membrane Transporter Expression Protocols in *E. coli*

Since *E. coli* can potentially lead to the production of milligrams of expressed proteins at low costs for both structural and functional studies, many different strategies/solutions have been conceived for overcoming the scarce propensity of bacteria to produce human membrane proteins.

### 3.1. Strain Selection

The amount of expressed protein depends on several factors such as the protein localization, the redox state of the protein, the sensitivity to bacterial protease, and the transcription/translation rate. The last factor could, in turn, be influenced by growth temperature, pH, osmotic pressure, or culture medium. Some features of the expressed protein, such as size, number of TM domains, presence of disulfides, and the presence of cleavage site or tags, can also influence the state of the protein or the tolerance by bacteria. Since an *E. coli* strain with supernatural powers, capable of expressing every desired protein, does not exist, a plethora of different strains has been developed, and these are commercially available for specific cultivation needs. Table 1 summarizes the most interesting strains for membrane protein expression, with the specific genetic modifications and features. Depending on target protein peculiarity, the best *E. coli* strain can be selected among this big portfolio. Indeed, starting from the most-used BL21, different modifications have been introduced for achieving novel properties, such as human tRNA supply (BL21 CodonPlus and derivatives, Rosetta); the induction of disulfide bridge formation (Origami); protease deficiency; cold shock chaperonin co-expression (Arctic); and increased mRNA stability (BL21 STAR). Moreover, spontaneous BL21 mutants competent in toxic protein production (C41, C43, Mt56, Evo21) have also been selected [41,42,43]. *E. coli* strain selection is crucial for protein production, especially in the case of membrane transporters. Specific features of the strains that have to be taken into consideration for the right choice are described in the following paragraphs.

#### 3.1.1. (DE3) or Non-DE3?

Several *E. coli* strains integrate the λDE3 lysogen, which carries the gene coding the T7 RNA polymerase under the control of the IPTG-sensitive *lac*UV5 promoter. When a DE3 strain is transformed with a recombinant plasmid in which the target gene is under the control of the T7 promoter, such as pET series vectors, after IPTG addition, the *lac*UV5 is activated and the T7 RNA polymerase is produced. The new synthesized T7 RNA polymerase, starts the transcription of the target gene since IPTG also activates the plasmidic T7 promoter. This mechanism maintains tight expression control. Indeed, in the absence of IPTG, the target protein cannot be expressed since the T7 RNA polymerase is not produced, the canonical bacterial RNA polymerase does not recognize/bind the T7 promoter that, in any case, is repressed. This is crucial since the leaky expression of a human hydrophobic protein, such as a transporter, can be toxic for bacteria [75]. The T7 RNA polymerase is more than five times faster than the endogenous *E. coli* RNA polymerase [41,76]. The rate of RNA synthesis must be well controlled since faster transcription can be useful in some cases, but can be deleterious in others. Indeed, the presence of naked RNA caused by faster RNA synthesis can lead to a decrease in protein expression [77]. Moreover, in the case of membrane protein expression, a fast transcription rate can result in Sec-translocon saturation, compromising cellular viability [78,79]. When the target gene is cloned under the control of the trp/lac hybrid promoter (tac), non-DE3 lysogen strains can be employed as hosts.

#### 3.1.2. pLysS, pLysE and Lemo21 Hosts

To prevent or reduce the toxicity of the target gene as it occurs when expressing human membrane transporters, tuning the RNA synthesis rate could provide benefits. This approach can be pursued using pLysS or pLysE strains containing a chloramphenicol-resistant plasmid expressing the T7 lysozyme. This is a natural inhibitor of T7 RNA polymerase, whose activity can be tuned by the inhibitor. This strategy allows the expression of relatively toxic genes in the same cell under the control of a T7 promoter [80,81]. The difference between the pLysS and pLysE hosts is the level of accumulated lysozyme, which is much higher in pLysE, causing a stronger decrease in growth rate with respect to the pLysS strain. The combination of the T7lac promoter and pLysS can be useful for very toxic genes. An alternative strategy for regulating the T7 RNA polymerase level is employed in the Lemo21 strain, in which the T7 lysozyme gene (inhibitor of the T7 RNAP) is inserted in a pLemo plasmid under the control of a rhamnose-inducible promoter. In this strain, both the synthesis and the activity of the T7 RNA polymerase can be tuned by rhamnose and IPTG levels. This, in turn, can regulate the production of the potential target gene [82]. Another advantage of the pLys strain in expressing membrane proteins is the lytic activity on the bacterial cell wall. This feature facilitates the solubilization of hydrophobic over-expressed proteins with non-ionic detergents such as Triton X-100.

#### 3.1.3. Rare tRNA Supplementation/Codon Usage Bias

Except for Methionine (AUG) and Tryptophan (UGG), each amino acid is coded by more than one codon. Each species “prefers” some of these codons based on its tRNA pools. This means that, in each cell, the tRNA population is closely related to the codon used by the mRNAs. When the mRNA of a human gene is expressed in *E. coli*, the differences in codon usage can impair translation due to the need for one or more tRNAs that may be rare or lacking in the host. Insufficient tRNA pools can lead to amino acid misincorporation, ribosomal stalling, premature translation termination, and translation frameshifting [83,84,85]. In particular, the least-used codons in *E. coli* are AGG, AGA, CGA, CGG, coding for Arginine, AUA coding for Isoleucine, CUA coding for Leucine, CCC coding for Proline, and UCG coding for Serine [86,87]. Several studies have reported a significant increase in protein yield when *E. coli* hosts are implemented with *argU* (AGG/AGA), *gly*T (GGA), *ile*X (AUA), *leu*W (CUA), or *pro*L (CCC) (see Table 1), which increase the respective tRNA levels [29,46,60,61,62,88]. Rosetta strains are added with tRNAs for the codons AGG, AGA, AUA, CUA, CCC, and GGA on a compatible chloramphenicol-resistant plasmid. A seventh rare codon (CGG) is present in the Rosetta 2 strain in addition to the six found in the Rosetta strain. BL21-CodonPlus-RIL contains tRNAs for the codons AGA, AGG, AUA, and CUA. This strain has been successfully used for the expression of several mitochondrial and plasma membrane transporters, which will be reported in detail in Section 4. The use of *E. coli* strains with tRNA supplementation is a low-cost alternative to the more expensive optimization of the entire cDNA. Several companies offer a complete gene synthesis service. This results not only in codon usage bias according to the genome of the host, but also in harmonizing the GC or CpG dinucleotide percentage, reducing mRNA secondary structure, and eliminating cryptic splicing sites, premature PolyA, internal chi sites (causing recombination), alternative ribosomal binding sites (causing wrong translation) and RNA instability motifs (ARE). Another factor potentially influencing expression in *E. coli* is the second codon. AAA (Lys) and AAT (Asn), are the two most frequent second codons in *E. coli* [89]. Therefore, changing the second codon in human cDNA may have beneficial effects on expression. A recently described aspect that can be taken into consideration is the nucleotide composition at 5′ End of the target gene. The first 18 nucleotides in the coding sequence, which are physically protected by the ribosome in the 70S initiation complex, have a strong influence on protein expression. At the beginning of genes, G reduces and A increases the probability of high expression, whereas C and U have intermediate effects; moreover, mRNA structure, and not codon usage, primarily determines the translation rate [90,91,92]. However, specific data on the first 18 nucleotide optimization on membrane proteins are still not available.

#### 3.1.4. Isolation of *E. coli* Mutants for Toxic Protein Production (C41, C43, Mt56)

Over-expression of membrane protein in *E. coli* can be toxic, killing the bacterial host. In 1996, Miroux and Walker transformed BL21(DE3) cells with cDNA coding for a mitochondrial oxoglutarate–malate carrier. After carrying out a specific cultivation protocol, they plated the corresponding mix on two sets of agar plates, one containing IPTG and the other lacking the inducer. In the presence of IPTG, two populations of bacteria differing in size (small and large) were obtained. Large colonies did not exhibit expression of the target protein, whereas the small ones produced the target protein at a high rate. The strain able to perform the toxic protein expression was defined C41(DE3) [41]. Starting from this newly isolated strain, the same protocol was used to isolate an additional strain, namely C43(DE3) [41]. Sequencing the complete genome of C41(DE3) revealed that the strain had acquired mutations in four different regions during the selection process [93]. In particular, a mutation in the P_lac UV5_ promoter occurred, drastically reducing its strength (P_lacWweak_) and causing a reduction in T7 RNA polymerase accumulation levels; this presented a very efficient way to alleviate protein production stress [93]. Using the same approach, another BL21(DE3) mutant strain enabled in membrane protein expression was developed and defined as Mt56 [42]. Sequencing of C43(DE3) evidenced mutations in the gene encoding *Lac*I, which is adjacent to T7 RNA polymerase; it was proposed that the mutated *Lac*I could bind more tightly to the lac operator, resulting in even tighter regulation of T7 RNA polymerase expression [94]. Sequencing of the Mt56 genome revealed one mutation in the gene encoding the T7 RNAP, and another in fryA gene that was predicted to be part of the putative PTS permease FryABC [95]. The point mutation of the gene encoding the T7 RNAP (A102D) occured at the site of the polymerase involved in T7 promoter binding, lowering its affinity towards the T7 promoter. Compared to BL21, as observed for C41 and C43, and also for the Mt56 strain, membrane protein production levels were reduced, lowering the saturation of the membrane protein biogenesis pathways and metabolic stress; this led to increased membrane protein expression on the cytoplasmic membrane [42].

#### 3.1.5. mRNA Stabilization

Increasing mRNA concentration could increase translation efficiency, resulting in an increase in protein production, even though it is reported that in *E. coli* and *L. lactis*, mRNA stability decreases with increasing concentration [96]. Another factor influencing mRNA stability (half-life) is the presence of the RNase E which is coded by *rne* gene, an endoribonuclease involved in most aspects of RNA processing and degradation in many bacteria [97]. In order to improve protein expression, an *E. coli* strain in which a mutant for of the *rne* gene lacks the C-terminal domain was developed (BL21Star). In the same scenario, by using the approach already used for toxic proteins [40,41,92], a new mutated BL21-derived strain, namely Evo21(DE3), was developed, harboring a truncated *rne* gene. This strain is, thus, similar to the BL21Star, but contains an additional mutation (D346N) in the N-terminal endoribonucleolytic domain of RNase E. Evo21(DE3) was able to express the membrane protein yidC [42].

### 3.2. Plasmid Selection

In addition to strain selection, the choice of the right plasmid is also crucial in membrane protein expression and purification. Several key sequences that can strongly influence protein production are distributed in the plasmid at 4–8 Kbp, such as the origin of replication, the translation initiation region, the promoter, the type and position of the tag, the antibiotic selection marker, and the terminator [30].

#### 3.2.1. Origin of Replication (Ori)

A single point mutation of the replication origin (ori) can drastically change a plasmid’s copy number [98]. The plasmid copy number (PCN) is <5 for pSC101 ori, and increases to 15–20 copies for pColE1 and pBR322 oris, which are present in the most common expression vectors such as pET and pGEX (See Table 2). In the case of pMB1, the PCN ranges from 15 to 60. A derivative of pMB1 is present in the pUC vectors, allowing an increase in the PCN to 500–700 [30]. Very recently, using fluorescent reporters, a method for simultaneously monitoring plasmids and RNA transcript copies in individual cells and protein expression was developed [99]. Interestingly, it was observed that the distribution of the plasmid copy number was wide, with mean copy numbers of 4, 11, 15 and 60 for pSC101, p15A, pColE1, and pUC, respectively. Moreover, a small fraction of cells did not contain plasmids, whereas many cells contained 4-fold or more of the mean copy number [99]. Although a high copy number of plasmids results in multiple copies of the gene of interest, it does not automatically result in higher protein expression, especially in the case of hydrophobic proteins such as membrane transport systems. In fact, high copy number plasmids, besides causing a metabolic burden on bacterial cells, may also induce aggregation of the produced protein if it is very hydrophobic. This will result in reduced growth rate and plasmid instability, thereby impairing the overall protein yield [100].

#### 3.2.2. The Promoter

The promoter is an important target for improving protein expression as it plays a key role in transcription regulation of the gene of interest. In general, the promoter should be strong and tightly regulated, with a minimum or complete lack of basal-level transcription in the absence of an inducer. Avoiding basal transcription is crucial for the expression of toxic proteins as their leaky expression can inhibit cell growth and reduce overall yield [47]. The strength of a promoter depends on the conservation of the canonical consensus sequences at the -10 and -35 positions. One of the better-studied promoters is the weak promoter lac, inducible by IPTG (in vitro) [115]. The induction of this promoter is maximal in the absence of glucose, which inhibits lactose uptake mediated by lac permease [49]. At low glucose levels, cAMP is produced with full activation of the lac operon [48]. Alternative promoters have also been used for expressing membrane proteins, such as the *lac*UV5, which is characterized by “leakiness” [116]; the synthetic tac [117] (present in pMAL series of vector); and trc promoters [118]. The last two promoters are strong and regulated by catabolite repression, allowing the accumulation of up to 15–30% of total cell proteins [119]. The T7 promoter system is one of the most widely used for protein expression, and is present in the pET vectors [76,120]. In case of toxicity of the target gene, such as for membrane transport systems, promoters with a lac operator sequence just downstream of the T7 promoter can be used (T7lac promoter). Plasmids containing T7lac also carry the natural promoter and coding sequence for the lac repressor (lacI), oriented so that the T7lac and lacI promoters diverge. When the use of this kind of plasmid is coupled with DE3 lysogens strains (see Section 3.1), the lac repressor acts both at the *lac*UV5 promoter in the host chromosome and at the T7lac promoter in the vector. This results in tight control of the transcription level, preventing leaky expression, which is very important in the case of hydrophobic proteins. This promoter/strain combination has been largely and successfully adopted for the expression of membrane transporters [44,46,50,52,56,66,67,69,121,122,123,124] (and see Table 3 and Section 4). Another approach could be used that consists of cloning the target gene under the control of the phage promoter pL, which has a moderately high expression level and is regulated by the cI repressor [125]. In this case, expression can be induced by changing the growth temperature from 30 °C to 42 °C. The temperature switch triggers protein expression by inactivating the temperature-sensitive cI857 repressor.

### 3.3. Cell-Free Protein Expression

An alternative strategy for avoiding membrane protein aggregation is the cell free expression system derived from *E. coli* extracts. The method, originally developed by Zubay in 1973 [134], has been improved in recent years for the expression of mammalian transporters [135,136,137,138], whose production in *E. coli* cells was virtually absent [139,140]. Transcription in the *E. coli* lysate depends on the strong T7 promoter and is facilitated by the addition of purified T7 RNA polymerase, disulfide kit, Mg^2+^, NTPs, and DNA template, either in the form of plasmidic DNA or as linear PCR products [141]. The translation process requires the addition of tRNA mixtures (human tRNA, using extract of BL21 CodonPlus RIL strain can be useful), the 20 amino acids, a ligand or a substrate of the target protein, and protease, and RNAse inhibitors. Moreover, in the case of membrane protein production, stabilizing agents such as lipids/liposomes or detergents can be added to the cell-free reaction mixture to trigger protein folding. Each of the parameters previously listed, also including growth temperature, *E. coli* strain, lipid composition, and detergent, have to be taken into consideration due to their influence on protein expression. Indeed, the addition of lipids or detergent was successfully adopted for the cell-free synthesis of a G protein–coupled receptor [142], two potassium channels [143,144], the human aquaglyceroporin 3 protein (AQP3) [145], three isoforms of the mitochondrial uncoupling protein (UCP1-3) [146], two isoforms of the mitochondrial ADP/ATP carrier (hAAC1 and hAAC3) [147], the human mitochondrial voltage-gated ion channel (VDAC) [148], and the human proton-couple folate transporter (PCFT) [149]. Interestingly, an *E. coli* cell-free expression system was successfully adopted for the production, purification, and crystallization of the human VDAC1 [150]. Taken together, these data highlight the possible alternative role of a cell-free expression system for the synthesis of reluctant-to-express proteins.

## 4. Successful Over-Expression of Transport Systems

Transporters belonging to the different groups described in Section 2 have been successfully over-expressed in bacteria. The following section will describe the crucial strategies used for each expressed protein, summarized in Table 3, to indicate solutions that could be adopted for many other proteins. The main features of transporters and transporter families and the major achievements obtained thanks to bacterial expression are also described.

### 4.1. SLC1

The SLC1 family includes seven members involved in amino acid traffic in cells. The family members are divided in two groups according to substrate specificity and transport mode [151]. The first group (members A1, A2, A3, A6, and A7), known as EAATs (excitatory amino acid transporters), includes transporters with high affinity for the negatively charged amino acids glutamate and aspartate; the second group includes SLC1A4 and A5, known as ASCTs (alanine, serine, and cysteine transporters), involved in the traffic of several neutral amino acids in a broad set of tissues [152]. The members of the SLC1 family possess eight membrane-spanning domains, are glycosylated, and contain between 524 and 574 amino acids. Even though the 3D-structures of the last two members have recently been solved by Cryo-EM using *P. pastoris* as the expression host [15,153], the ASCT2 protein was also produced in *E. coli* [53]. The expression strategy employed the use of the Rosetta-gami2 strain, which joins the human tRNA supply (Section 3.1) and is involved in disulfide formation (Section 3). For successful expression, a low temperature strategy had to be used to reduce protein toxicity. To achieve this, the pCOLD I vector—which carries the cold shock protein (csp), a promoter activated at low temperature—was used for cDNA cloning. Before the addition of 0.4 mM IPTG, a cold shock (10 min on ice and 40 min at 15 °C) was performed to improve the transcription of the ASCT2 mRNA and the stability of the 5′-UTR according to the feature of the cold-shock protein A promoter [154]. To antagonize the protein toxicity observed as cell culture OD reduction, glucose was added to prevent leakiness, and the growth temperature post-induction was kept at 15 °C to reduce the basal metabolism.

### 4.2. SLC2

The SLC2 family includes 14 GLUT members involved in the transport of monosaccharides, polyols, and other small carbon compounds across eukaryotic cell membranes [155]. The GLUT proteins contain about 500 amino acid residues, are glycosylated, and have 12 membrane-spanning domains. The human GLUT1 transporter, coded by the SLC2A1 gene, is the first example of a human transporter expressed in *E. coli* [74]. The cDNA coding for hGLUT1 was cloned into a pGTSD12 plasmid with an upstream prokaryote-type ribosome binding site in a T7 promoter/T7 polymerase expression system. The strategy consisted of exploiting the insertion of the recombinant GLUT1 protein in the cell membrane of the SR425 *E. coli* strain, which was rid of all the bacterial glucose transporter genes (*pts*G, *ptsM*, *gal*) and transformed with the gene coding for the T7 RNA polymerase under the control of the heat-inducible pL promoter. In this case, the transport function of hGLUT1 could be assayed directly in the bacterial host. ^14^C-glucose transport inhibited by 2-deoxy-D-glucose and D-glucose, but not by L-glucose, was observed, confirming the expression of the target protein. In this system, the kinetics of transport were also studied, highlighting that—as observed in erythrocytes—glucose transport is inhibited by cytochalasin B and by mercuric chloride [74].

### 4.3. SLC3

The SLC3 family includes two members—SLC3A1 (also named rBAT) and SLC3A2 (also named 4F2hc or CD98hc)—that share about 20% of their amino acid sequence identity [156]. Both proteins are N-glycosylated: ~94 and ~85 kDa for the mature glycosylated forms of rBAT (685 aa) and 4F2hc (630 aa), respectively. The two proteins are type-II membrane N-glycoproteins with a single TMD and an intracellular N-terminus. They are characterized by a bulky extracellular C-terminus domain (50–60 kDa) that has been expressed in *E. coli*. Being highly water soluble, this domain has been crystallized and its structure solved by X-ray diffraction [157]. Even though these proteins are classified as members of the SLC superfamily, they are not directly involved in solute transport, but form heterodimers with some members of the SLC7 family, which are the subunits competent for transport [158]. cDNA coding for the entirety of SLC3A2 was cloned in a pGEX-4T1 vector that includes an N-terminal GST tag. The corresponding protein could then be over-expressed in Rosetta(DE3)pLysS, probably due to the higher solubility of the chimeric protein (SLC3A2-GST) with respect to the sole SLC3A2. The chimeric protein was purified on a glutathione Sepharose 4B affinity column, then cleaved using thrombin treatment [61].

### 4.4. SLC5

The SLC5 family includes 12 members that are sodium-dependent transporters involved in intestinal absorption (SLC5A1) or in the renal re-absorption (SLC5A2) of sugars [159]. The sole member of this family expressed in *E. coli* is SLC5A1 [126]. The use of a BL21 strain defective in the outer membrane protease (OmpT), together with low incubation temperatures (16 °C) and transcriptional regulation from the lac promoter/operator, have been crucial to reducing proteolytic degradation. Because bacterial cotransporters possess significantly shorter N-terminal hydrophilic extensions with respect to their eukaryote counterparts [160], amino acid residues 12–28 were removed, promoting insertion in the *E. coli* membrane. To recover the over-expressed protein and assay the transport function, SLC5A1 was solubilized with FosCholine-12 detergent, purified, and reconstituted in proteoliposomes in an active form [126].

### 4.5. SLC6

This family includes 20 secondary active co-transporters with 12 membrane-spanning domains that utilize a chemiosmotic Na^+^ gradient and/or Cl^−^ to couple the transport of their substrates across a membrane [161]. The serotonin transporter (SLC6A4) was expressed and targeted to *E. coli* membranes by combining codon optimization and tRNA supply in different strains and media [57]. Another member of the SLC6 family, the SLC6A19 (also named B0AT1), was expressed in *E. coli* exploiting human tRNA supplementation (BL21 CodonPlus RIL strain), combining a cold shock strategy (csp promoter, see SLC1A5) with a very low inducer concentration (10 μM) in the presence of 0.5% glucose [53]. This protein is in complex with ACE2, constituting part of the receptor for the SARS-CoV-2 RBD proteins [162]. The production of B0AT1 in *E. coli* at a high yield can be useful for studying the interaction with compounds, which may have the potential for application as COVID-19 drugs [163,164].

### 4.6. SLC7

The SLC7 family includes 13 members divided in two subfamilies: the cationic amino acid transporters (CATs, SLC7A1–4, and SLC7A14), and the light or catalytic subunits (L-type amino acid transporters LATs, and SLC7A5-13) of the heteromeric amino acid transporters (HATs); these are mostly exchangers with a broad spectrum of substrates, ranging from neutral to negatively charged amino acids [156]. The members of this family differ in length—ranging from 470 amino acids for the SLC7A13 to 771 amino acids for the SLC7A14 members—and, consequently, in TM domains (12–14). SLC7A5 was over-expressed in the Rosetta(DE3)pLysS strain, (human tRNA supply) under standard conditions such as 4 h of 0.4 mM IPTG induction at 28 °C [61]. The addition of an N-terminal 6His tag was crucial for IMAC purification of a protein that was refolded on a column and reconstituted into proteoliposomes. Several structure/function relationships were also defined thanks to the production of several mutants, with relevance to physiology and pathology [165,166,167].

### 4.7. SLC17

The SLC17 family is a group of nine structurally related membrane proteins that mediate the transport of organic anions. SLC17A1–4s, also known as type-I phosphate transporters, are involved in the sodium-dependent transport of inorganic phosphate and other organic anions, such as urate and para-aminohippurate [168]. The member SLC17A5, also known as sialin, catalyzes the lysosomal transport of sialic acid and acidic sugar, including glucuronic acid. The SLC17A6-8s (vGLUTs) localize to synaptic vesicles, but each appears to have a different distribution among other cell membranes where they are involved in glutamate transport [169]. The SLC17A9 member (VNUT) is expressed in several tissues in mammals and is involved in vesicular nucleotide transport [170]. The SLC17 family members share a similar topology, since they are predicted to have 12 TM domains with intracellular N- and C-*termini*, as confirmed by the recently solved structure of the rat orthologous SLC17A6 [171]. Two out of nine members of the SLC17 family were expressed in *E. coli*, exploiting a strategy that combines low temperature expression and membrane targeting using N-terminal and/or C-terminal fusion peptides, constituting 120 amino acids of the YbeL bacterial protein (named β- domain) [66]. In particular, β- domains were added at both the N- and C-termini of the SLC17A5 and only at the C terminus of the SLC17A9 cDNAs. The fusion constructs were cloned in a pET-28a(+) vector. C43 cells were induced with 1 mM IPTG at 18 °C for 16 h in order to promote the insertion of the proteins in the membrane of the bacterial host. The proteins were purified using Ni-NTA chromatography and reconstituted in liposomes in a functionally active state [66].

### 4.8. SLC18

The four members of the SLC18 family mediate the transport of neurotransmitters (SLC18A1-A3) or polyamines (SLC18B1) [172,173]. Most computer-based predictions suggest 12 TMs with the N- and C-termini facing the intracellular milieu and a larger luminal loop between the first and second transmembrane domains. For the expression of the SLC18A3/VAChT, which mediates the transport of acetylcholine, the same strategy adopted for the members of the SLC17 family was used. The insertion in the C43(DE3) membrane was triggered by adding a YbeL tag at both the N- and C-termini of the protein after cloning in a pET-28a(+) vector. The induction of protein synthesis occurred at 18 °C for 16 h in the presence of 1 mM IPTG [20]. The use of a 6His tag allowed purification via Ni-NTA affinity chromatography, followed by MALDI mass spectrometry analysis. Which confirmed the production of the target protein [20].

### 4.9. SLC22

The SLC22 family is consists of at least 31 transporters expressed on both the apical and basolateral surfaces of epithelial cells, where they direct small-molecule transport between the body fluids and vital organs, such as the kidney, liver, heart, and brain [174,175]. The family includes organic cation transporters (OCTs), novel organic cation transporters (OCTNs), and organic anion transporters (OATs) with different modes of transport. They have been defined as “drug transporters” due to their role in the absorption and excretion of drugs. These proteins share 12 α-helical TM domains, a large extracellular domain between TM1 and TM2, and a large intracellular domain between TM6 and TM7 [176]. Two members of the OCTN subfamily (SLC22A4 and SLC22A5) were over-expressed in *E. coli* with different strategies [46,62]. The cDNA coding for SLC22A4/OCTN1 was cloned in the pH6EX3 expression vector, and the Rosetta(DE3)pLysS strain was used for human tRNA supply. Protein synthesis was induced with 0.4 mM IPTG at 28 °C for 6 h and a 6His-tagged protein was purified and functionally reconstituted in proteoliposomes [62,177,178]. After codon optimization, the use of the Lemo21 strain strongly increased the production of the target protein [127]. The over-expression of OCTN1 and the production of several mutants allowed researchers to reveal the structure/function relationships between this transporter and the molecular basis of human diseases [177,178,179]. Moreover, a specific antibody (anti-OCTN1) was produced using the over-expressed protein [180]. Despite the high sequence similarity (86.5%), the same approach was not effective for SLC22A5/OCTN2. For the production of the protein, the use of an N-terminal GST tag was exploited, but the amount of protein recovered after expression in Rosetta(DE3)pLysS and tag removal was quite low [46]. Interestingly, the conservative substitution of the second codon (R2K), introducing the statistically most present codon at the second position of *E. coli* genes [89], allowed the production of the target protein in a much larger amount with respect to the wild-type protein [46].

### 4.10. SLC25

With its 53 members, the SLC25 family is the largest among the solute carrier families. Based on sequence similarities, human MCs cluster into many different clades, suggesting a large variety of transported substrates, among which are nucleotides, carboxylates, amino acids [181]. Most MCs contain about 300 amino acids, with six TMs and the N- and C-termini protruding towards the cytosolic side of the inner membrane. For the expression of several members of the SLC25 family, neither human tRNA supply nor codon optimization was necessary (Table 3). The most-used strategy for the expression of several members of the SLC25 family was based on inclusion body formation, obtained by combining a BL21(DE3) derivative strain with a pET or T7 derivative plasmid (Table 3). In particular, the first protein over-expressed in a bacterial host, and then purified and assayed in an in vitro system, was the bovine oxoglutarate carrier, which was cloned in a pRUN vector [114]. Then, the oxodicarboxylate (ODC, SLC25A21), and glutamate carrier 1 and 2 (SLC25A22 and SLC25A18, respectively) were also cloned in the pRUN plasmid. The corresponding proteins were expressed as inclusion bodies after culturing C0214 cells for 4.5 h in the presence of 0.4 mM IPTG at 37 °C, and reconstituted in proteoliposomes in an active form [68,69]. The same method with a similar plasmid or strain was used for studying the carnitine/acylcarnitine translocase (CACT) [67]; the ornithine/citrulline carrier (ORNT1) [128]; the basic amino acid transporter (SLC25A29/ORNT3) [129]; the S-adenosylmethionine transporter (SLC25A26) [54]; the ATP-Mg/Pi transporters APC1 (SLC25A24) and APC3 (SLC25A25) [72]; and the peroxisomal transporter of coenzyme A, FAD and NAD^+^ (SLC25A17/PMP34) [63]. This strategy was also used for the production of all the UCP isoforms (see Table 3). Indeed, Ivanova et al. cloned all five UCPs in pET-21a(+) and expressed the corresponding proteins in BL21(DE3) as a 6His-tagged protein in inclusion bodies, after 3h of 1 mM IPTG induction at 37 °C [56]. Following purification, the UCPs were reconstituted in stable, small, unilamellar vesicles, as confirmed by circular dichroism analysis [56]. The same expression strategy was adopted for the expression of UCP2 and UCP3, which were refolded from inclusion bodies using a dialysis method [65,123]. A completely different approach was used for the expression of the UCP1 protein, in which the insertion of the protein into the *E. coli* membrane was obtained by adding the small periplasmic leader sequence (PelB), provided by the pET-26b(+) vector and culturing BL21 CodonPlus(DE3)-RIPL with an auto-induction method [55]. The protein was extracted from the *E. coli* membrane, purified, and reconstituted into phospholipid bilayers in an active form [182]. In addition to *E. coli*, *L. lactis* was also used for the expression of the regulatory domains of aralar1 (SLC25A12) and citrin (SLC25A13). In particular, the N- and C-terminal domain of both proteins were cloned in the pNZ8048 expression vector, produced as 8His-tagged proteins in *L. lactis* NZ9000, purified using a nickel-Sepharose high-performance column, and crystallized [13].

### 4.11. SLC29

The SLC29 family includes four members present in most cell types and tissue, designated equilibrative nucleoside transporters (ENTs) because of the properties of the first-characterised member of the family, hENT1. ENT1 and ENT2, have a similar broad substrate specificity for purine and pyrimidine nucleosides, but ENT2 is also competent in nucleobases transport [183]. ENT1 and ENT2 are localized in the basolateral membrane of polarized cells. ENT3 is widely distributed and is present in lysosomal and mitochondrial membranes [184]. ENT4 is present on plasma membranes and transports adenosine and monoamines to the brain and heart [185]. The bacterial expression of SLC29A1/ENT1 tagged with an N-HAT-3X-FLAG was obtained in BL21(DE3) under the control of a lac promoter. The use of this promoter was crucial, since it allowed a slow constitutive transcription rate that was well tolerated by the translational machinery of the bacterial host when cultured below 25 °C [107].

### 4.12. SLC30

The SLC30 family includes 10 members involved in Zn^2+^ homeostasis in the body. Based on sequence similarities, the SLC30 family members can be divided into four subfamilies. Despite the difference in amino acid length—ranging from 323 amino acids for SLC30A2 to 765 amino acids for SLC30A5—all the members of the family are predicted to have six TM domains and a histidine/serine-rich loop between TM4 and TM5. The N- and C-terminal ends are localized intracellularly, facing the cytosol [186]. A member of this family, ZnT8 (SLC30A8)—and in particular, the C-terminal domain of the transporter (amino acids 268–369)—was identified as a target of autoantibodies in type-I Diabetes Mellitus (DM) [187]. These auto-antibodies were detected in more than 60% of patients with type 1 DM [187]. Thus, the over-expression of the C-terminal domain can help in the diagnosis of type-I DM. The codon-optimized sequence coding for the C-terminal domain of the ZnT8 transporter was cloned in a specific plasmid and the corresponding peptide was produced as thioredoxin (Trx), tagged either in inclusion bodies using the GI724 strain or in the soluble fraction of cell lysate culturing GI698 cells (Table 3). The peptide was successfully purified in a properly folded state that could be useful for developing new low-cost tests for the detection of ZnT8 autoantibodies [45].

### 4.13. SLC35

The SLC35 family includes thirty-one members that are divided into seven subfamilies, from SLC35A to SLC35G, also named nucleotide sugar transporters (NST); they connect the synthesis of activated sugars in the nucleus or cytosol to glycosyltransferases that reside in endoplasmic reticulum (ER) and/or Golgi apparatus [188]. This connection is crucial for regulating sugar- and organelle-specific protein glycosylation [189], and there are further indications that some members of this family may also form multiprotein complexes with glycosyltransferases [190]. For the majority of the SLC35 family members, a topology characterized by 10 TM α-helices connected by short loops, with N- and C-termini located on the cytoplasmic side, is predicted [189]. The expression of the GDP-L-fucose antiporter SLC35C1 in an active form was obtained in the BW25113(DE3) *E. coli* strain specifically deleted (*∆fucAK)* (see Table 1) [71]. The expression strategy combined the codon optimization of the gene and the use of a N-terminal OmpA signal sequence, promoting the insertion of the target protein in the bacterial membrane. The cultures were induced for five hours with 0.01% L-rhamnose. To study the GDP-L-fucose transport *in vitro*, inside-out vesicles of the OmpA-FucT1 expressing *E. coli* strains were produced and kinetic parameters were also measured, confirming the production of the target protein in a functional state [71]. Another member of the SLC35 family, (the putative thiamine transporter, member F3), was successfully produced in *E. coli* [51]. The plasmid pZE12luc with a strong and tightly regulated P_L_lacO1 promoter was used for cloning the target gene [191]. The MW25113 strain with either thiamine transport [ThiBPQ = sfuABC] or with its biosynthetic [*ThiH*] genes inactivated was used for producing the transporter in the bacterial membrane. The uptake of [^3^H]-thiamine confirmed the expression of the protein in a folded conformation and allowed researchers to perform functional studies [51].

### 4.14. SLC38

The SLC38 family includes eleven members and is part of a phylogenetic cluster of amino acid transporters comprising the SLC32 and SLC36 families [192]. They mediate the Na^+^-dependent net uptake and efflux of small neutral amino acids. They are particularly expressed in cells that grow actively, or in cells with active amino acid metabolism, such as the liver, kidney, and brain [193]. The overall structure most likely consists of 11 (TM) α-helices with the N-terminal domain protruding towards the cytosol. One of the most interesting members of this family is A9, which is a lysosomal amino acid transporter involved in sensing amino acid sufficiency for the activation of the mechanistic target of rapamycin complex 1 (mTORC1), which regulates cell growth and proliferation [70,194]. The heterologous expression of the A9 member of the SLC38 family in *E. coli* caused cell death, probably as a consequence of Sec-translocon saturation. To overcome this problem, a codon optimization strategy with modulation of RNA polymerase T7 activity was attempted [70]. Culturing a Lemo21(DE3) strain in the presence of 0.1 mM rhamnose to induce lysozyme T7 production (see Section 3.2), but for only 2 h at 39 °C, allowed the synthesis of the target protein, which was reconstituted in proteoliposomes in an active form [70,195].

### 4.15. SLC52

The SLC52 family includes three members (SLC52A1-3) involved in regulating riboflavin homeostasis. SLC52A1 is highly expressed in the placenta and small intestine, SLC52A2 is ubiquitous, and SLC52A3 is predominantly expressed in the testes and small intestine [196]. Riboflavin is the precursor of the flavin mononucleotide (FMN) and flavin adenine dinucleotide (FAD), which are essential cofactors for numerous enzymes (i.e., oxido-reductases, monooxygenases, oxidases); thus, the alteration of the flavoproteome can cause severe diseases [197]. Hydropathy profile analysis predicts 11 TM domains for all members of the family. To express member 2 of this family (SLC52A2), its codon-optimized cDNA was cloned under the control of the strong tac promoter, and the production of the corresponding 6His-tagged protein was obtained by culturing the Rosetta(DE3) strain at 28 °C in the presence of 0.4 mM IPTG [60]. The protein was purified by nickel-chelating affinity chromatography and reconstituted in proteoliposomes in a folded state; this allowed the discovery of novel aspects of the transport mechanism, such as inhibition by FMN and the interaction with Ca^2+^ ions [60]. Moreover, structure/function relationships were characterized thanks to the bacterially expressed proteins [59,60].

### 4.16. ABCB10

A small number of ATP-binding cassette (ABC) transporters are found in mitochondria. Most are half-transporters of the B-group-forming homodimers, and their topology suggests that they function as exporters [198]. The proteins contain an N-terminal transmembrane domain (TMD) and a C-terminal nucleotide-binding domain (NBD), which form homodimers for full functionality. To obtain the bacterial expression of the ABCB10 protein, the key factors were the use of a codon-optimized sequence and an N-terminal 6His tag obtained after cloning in the pET-19b vector. The cultivation of Rosetta 2, BL21-CodonPlus(DE3)-RIPL or Lemo21(DE3) gave similar positive results, allowing the production of the protein of interest after 3 h of induction at 30 °C in presence of 0.5 mM IPTG. The 6His-ABCB10 protein was solubilized from *E. coli* membranes and purified using Ni-NTA affinity chromatography. The folded state of the purified protein was assessed by circular dichroism spectra and reconstituted into a small lipid-bilayer system known as nanodiscs [52].

### 4.17. MDR1/P-gp

MDR1, also known as P-glycoprotein (P-gp), is a member of the ABC superfamily, present in both prokaryotes and eukaryotes, which regulate the traffic of various molecules and the extrusion of xenobiotics from cells. This 170-kDa membrane protein confers cross-resistance to many natural drugs such as anthracyclines, actinomycin D, Vinca alkaloids, and peptide antibiotics [199]. The strategy adopted for the bacterial expression of MDR1 was the use of the heat-inducible pL promoter. The full-length cDNA coding for MDR1 was cloned into a pPOW-B2 vector. UT5600 cells were cultured at 30 °C and the expression of the target protein was induced, shifting the temperature to 42 °C for 30 min. The production of the MDR1 protein in the *E. coli* membrane was confirmed by transport assays in which the uptake of ^3^H TPP^+^ (tetraphenylphosphonium) was measured [73]. Some years later, Linton and Higgins produced a lot of chimeric proteins of truncated P-gp fused to β-lactamase, to solve the enigma of the different topology when expressed in bacterial or mammalian cells. They concluded that P-gp could misfold during *E. coli* expression due to the misrecognition of multiple P-gp sequences as topogenic signals [200]. Altogether, the described works on ABC transporters indicate that, in the case of ABCB10, *E. coli* is suitable, likely due to its nature as a mitochondrial protein whose properties resemble those in bacteria. On the other hand, in other cases, we can expect that new technology would make *E. coli* a more favorable host.

### 4.18. TRP Proteins

TRP (transient receptor potential) proteins are cation-selective channels that become permeable in response to a wide variety of physical, chemical and thermal signals [201]. The TRP family consists of 28 members that form six-transmembrane (6-TM) channels which, on the basis of amino acid sequence homology, are divided into six subfamilies [202]. They assemble into tetrameric complexes, with their N- and C-terminals facing the cytoplasm. TRPV3 is a 91 kDa integral plasma membrane protein with six transmembrane helices and a pore region consisting of the ‘P-loop’. Heterologous expression of the GFP-tagged TRPV3 channel was obtained combining a tRNA supply and low temperature cultivation (See Table 3). In particular, protein synthesis was induced for 6 h at 18 °C in BL21(DE3)-R3-pRARE 2 cells. The inner membrane fraction of *E. coli* was obtained using sucrose gradient centrifugation, and after solubilization, the target protein was purified by IMAC. The transport activity and structure/function relationships were studied using black lipid membrane experiments, confirming the functional characteristics observed for TRPV3 expressed in human cells [64].

### 4.19. K^+^ Channels

Potassium channels (KChs) are membrane proteins involved in the regulation of many physiological processes, such as cell excitability [203], hormone secretion [204], proliferation [205], and apoptosis [206]. Even though they share a high similarity within the pore, known as the K^+^ selectivity filter, there is high divergence among the sensing domains, resulting in KCh gating in response to a wide variety of signals [207]. From a structural point of view, KChs are classified into four groups with different structure/topology and regulation: The members of the first group are tetramers characterized by six TMs with an intracellular N- and C- terminus. Among this group, there is the voltage-gated KCh (Kv) and the small and intermediate conductance Ca^2+^-activated KCh (KCa). The inward rectifier, KChs (Kir), the KATP, and the G-protein-coupled KChs are members of the second group. They are tetramers, formed by two TMs and one pore each. The large-conductance K_Ca_s are members of the third group and are characterized by seven TMs. Finally, the K_2P_ channels are dimers formed by four TMs and two pore regions [208]. For the expression of the intracellular domain of the EAG1 potassium channel (Kv10.1), a standard approach was adopted. The desired fragment was cloned in a high-copy plasmid and, following the transformation of the BL21(DE3) strain, the production of the 6His-tagged fragment was induced with 1 mM IPTG, at 37 °C for 4h. The protein was purified by nickel-chelating affinity chromatography and used for the production of a monoclonal antibody [130]. Quite a similar strategy was adopted for the expression of MiRP1 (MinK-related protein 1) an ancillary membrane protein of the KCNE family that associates with and modulates various voltage-gated potassium channels. The protein was produced using standard protocol (see Table 3), with the help of tRNA supply. Thanks to the C-terminal 6His tag added after cloning in a pET-21b plasmid, the protein was purified in different detergents and its folded state was confirmed using circular dichroism (CD) analysis [57]. A different approach was adopted for the expression of the ATP-sensitive inward rectifier potassium channel 1 (ROMK1). The codon-optimized cDNA was cloned in the pET-28(a)+ vector with different tags (MISTIC, OmpF, 6His, SUMO, Kir1.3Bac) for directing the target protein towards the *E. coli* membrane, or for improving the solubility. The BL21(DE3)-pLysS strain was used for the successful expression of the following constructs: ROMK1-6His, MISTIC-KirBac1.3-ROMK1, MISTIC-ROMK1, OmpF-KirBac1.3-ROMK1, and SUMO-ROMK1. MISTIC; moreover, MISTIC-KirBac1.3-tagged proteins were produced in pellet fraction, while ROMK1-6His protein was produced in the *E. coli* membrane. Using the 6His tag, the channel was purified and functionally inserted in planar lipid bilayers. The channel properties observed in these experiments correspond to those reported before for ROMK channels expressed in mammalian cells [124].

### 4.20. CLIC4

Chloride channels are located in the plasma membrane and other intracellular membranes (Golgi, endosomes, endoplasmic reticulum, and sarcoplasmic reticulum) where they participate in the control of secretion, the absorption of salts, the regulation of membrane potentials, organellar acidification, and cell volume homeostasis [209,210]. To date, seven members of the CLIC (chloride intracellular channel) characterized by a single TM domain have been studied. The wild-type cDNA coding for member 4 of this family (CLIC4) was cloned in pET-22b (+) and expressed in the BL21(DE3) strain as a C-terminal 6His-tagged protein following standard procedure (Table 3). After purification using on column Ni^2+^-chelating affinity chromatography, the protein was concentrated, crystallized, and solved using X-ray diffraction [131].

### 4.21. MICU1 and MICU2

MICU1 and MICU2 are protein subunits interacting with the mitochondrial calcium uniporter (MCU), and are involved in electrophoretic calcium transport in mitochondria [211]. The calcium-binding EF hand-containing proteins MICU1 and MICU2, associated with the pore complex in the intermembrane space, are the key regulators of uniporter signal processing [211]. Despite the clear role in mitochondrial calcium transport, the information about MICU localization is controversial, since MICU1 is referred to as a single-pass membrane protein of the mitochondrial inner membrane [212], or localized at mitochondrial intermembrane space [213]. Both MICU1 and MICU2 proteins were expressed in the BL21(DE3) *E. coli* strain using a low-temperature strategy (see Table 3). In particular, wild-type MICU1 was expressed with an N-terminal GST tag, whereas MICU2 was expressed in a truncated form (NΔ84- CΔ28) with an N-terminal 6His tag, and was recovered in the soluble fraction of the induced cell lysate [49]. After purification of MICU2 using Ni-NTA resin, the interaction between the GST-MICU1 and MICU2 was confirmed using GST pull-down analysis [50].

### 4.22. VDACs

Voltage-dependent anion-selective channels (VDACs), also known as mitochondrial porins, play the role of main regulators of metabolite exchange between the cytosol and mitochondria [214]. In humans, three isoforms have been described: VDAC1, VDAC2, and VDAC3. Despite the similarity in sequence and structure, evidence suggests different biological roles in normal and pathological conditions for each isoform [215]. Either wild-type VDAC1 or VDAC2 have been expressed as C-terminal 6His-tagged proteins in the *E. coli* M15(pREP4) strain in standard conditions (Table 3), purified by nickel-chelating chromatography, and reconstituted in planar bilayer membranes in a functional form [133]. Further structure/function relationship aspects of the VDAC1 protein were clarified after expression in BL21(DE3) under the standard conditions (see Table 3) of a deleted form (∆β9-∆β10). The C-terminal 6His-tagged deleted protein was purified and reconstituted in planar phospholipid membranes in a folded active state [121]. The same expression strategy used for the deleted VDAC1 was useful for producing VDAC3 and for demonstrating that this protein also possesses ion channel activity [44]. The 3D structure of VDAC1 was solved using NMR [132].

## 5. Conclusions

Expressing human membrane transport systems is mandatory for achieving systematic knowledge of this class of proteins. Due to the high expression capacity, low costs and well-known genetics and metabolism, bacteria—in particular, *E. coli*—represent the best compromise for producing a large number of human membrane transport systems. Using different in vitro experimental tools, the produced human transport proteins can be characterized in terms of function and structure/function relationships. Kinetics and the effects of specific regulators and inhibitors can be studied in a tightly controlled fashion, and specific amino acid resides crucial for transport can be identified. Concerning the suitability of bacterially expressed transporters for X-ray crystallography studies, this is not always satisfactory; this is due to the occurrence of protein-aggregation phenomena caused by the low solubility and/or a low tolerance for human proteins by bacteria, unlike bacterial transporters, whose structures were solved using bacterially expressed transporters. As an example, the lactose permease of *E. coli* (*lac*Y) was expressed in *E. coli* XL1-Blue cells, which were purified and crystallized [216]. The 3D structure of human transporters only started to be successful after the Cryo-EM revolution [217]. This technology required the expression of the transporters mainly by human hosts, with a huge increase in cost with respect to bacterial hosts. However, even after the 3D solution, the study of a transporter is far from being complete. Thus, *E. coli* will provide perspectives for future studies. Indeed, bacterial expression remains a powerful tool for further biochemical and structure/function characterization, also in terms of site-specific mutagenesis. Therefore, bacterial expression will be of great help in the coming years, to elucidate the molecular mechanism underlying the function of a transport system and the link of transporter defects with human pathology.

## Figures and Tables

**Figure 1 ijms-23-03823-f001:**
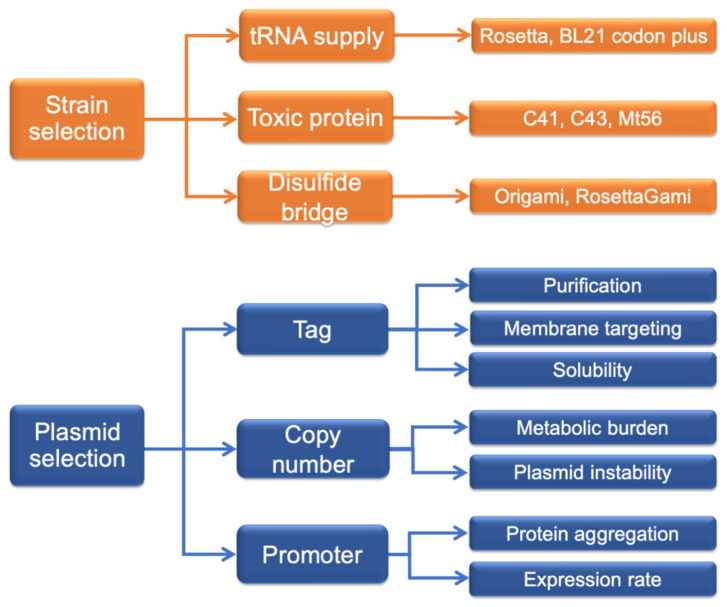
Graphical sketch of *E. coli* strains and plasmid features influencing protein production and state.

**Table 1 ijms-23-03823-t001:** *E. coli* strains used for human membrane transport systems expression and their main features.

Strain	Genotype	Feature	Reference
BL21(DE3)	*F^-^ ompT hsdS_B_,(r_B_^-^ m_B_^-^) gal dcm (DE3)*	T7 RNAP	[44,45,46,47,48,49,50,51]
Rosetta2	*F^-^ ompT hsdS_B_(r_B_^-^ m_B_^-^) gal dcm pRARE2 (Cam^R^)*	tRNA supply	[52]
BL21- CodonPlus-RIL	*E. coli B F^-^ ompT hsdS (r_B_^-^ m_B_^-^) dcm Tet^r^ gal endA Hte [argU ileY leuW Cam^r^]*	tRNA supply	[53]
BL21(DE3)-CodonPlus-RIL	*E. coli B F^-^ ompT hsdS (r_B_^-^ m_B_^-^) dcm Tet^r^ gal λ(DE3) endA Hte [argU ileY leuW Cam^r^]*	T7 RNAP, tRNA supply	[54]
BL21(DE3)-CodonPlus-RIPL	*E. coli B F^-^ ompT hsdS (r_B_^-^ m_B_^-^) dcm Tet^r^ gal λ(DE3) endA Hte [argU proL Cam^r^] [argU ileY leuW Strep/Spec^r^]*	T7 RNAP, tRNA supply	[52,55,56]
BL21(DE3)-CodonPlus-RP	*E. coli B F^-^ ompT hsdS (* *r_B_^-^ m_B_^-^* *) dcm Tet^r^ gal λ* *(DE3) endA Hte [argU proL Cam^r^]*	T7 RNAP, tRNA supply	[57,58]
Rosetta(DE3)	*F^-^ ompT hsdS_B_(r_B_^-^ m_B_^-^) gal dcm pRARE (Cam^R^)*	T7 RNAP, tRNA supply	[59,60]
Rosetta(DE3) pLysS	*F^-^ ompT hsdS_B_(r_B_^-^ m_B_^-^) gal dcm pLysSRARE (Cam^R^)*	T7 RNAP, tRNA supply, T7 Lysozyme	[46,61,62]
Rosetta-gami2	*Δara-leu7697 ΔlacX74 ΔphoA PvuII phoR araD139 ahpC galE galK rpsL* *F ’[lac^+^ lacI^q^ pro] gor522::Tn10 trxB pRARE2 (Cam^R^, Str^R^, Tet^R^)*	tRNA supply, disulfide formation	[53]
Rosetta-gamiB	*F^-^ ompT hsdS_B_(r_B_^-^ m_B_^-^) gal dcm lacY1 ahpC gor522::Tn10 trxB pRARE (Cam^R^, Str^R^, Tet^R^)*	tRNA supply, disulfide formation, lac permease mutation	[63]
BL21(DE3)-R3- pRARE 2	*F^-^ ompT hsdS_B_,(r_B_^-^ m_B_^-^) gal dcm (DE3) pRARE2 (Cam^R^)*	T7 RNAP, tRNA supply	[64]
C41	* F^-^ ompT hsdS_B_,(r_B_^-^ m_B_^-^) gal dcm *	BL21 mutant for toxic protein	[65]
C43(DE3)	*F^-^ ompT hsdS_B_,(r_B_^-^ m_B_^-^) gal dcm (DE3)*	BL21 mutant for toxic protein	[20,66]
C0214	* F^-^ ompT hsdS_B_,(r_B_^-^ m_B_^-^) gal dcm *	C41 mutant for toxic protein	[67,68,69]
Lemo21(DE3)	*fhuA2 [lon] ompT gal (λ* *DE3) [dcm] ∆hsdS/pLemo(Cam^R^)*	T7 RNAP, T7 lysozyme	[70]
*E. coli* BW25113(DE3) ΔfucAK	*∆(araD-araB)567, ∆lacZ4787(::rrnB-3), lambda-, rph-1, ∆(rhaD-rhaB)568, hsdR514, ∆(fucAK)*	T7 RNAP, fucose transport deletion	[71]
M15pREP4	*lac,ara,gal,mtl,recA^+^,uvr^+^[pREP4,lacI,kana^R^]*	lacZ mutation	[72]
MW25113	No info available	Endogenous thiamine transport, [ThiH] genes inactivated	[51]
UT5600	*F^-^ ara-14 leuB6 secA6 lacY1 proC14 tsx-67 Δ(ompT-fepC)266 entA403 trpE38 rfbD1 rpsL109 xyl-5 mtl-1 thi-1*	OmpT protease deficiency	[73]
SR425	*gal, thi, Tl^r^, endA, hsdR, sbcB, ptsM, ptsG/F^-^*	Defective for glucose transporter	[74]
GI724	*F^-^ lacI^q^ lacPL8 ampC::lambdacI^+^*	pL promoter,	[45]
GI698	*F^-^ λ* *-lacI^q^ lacPL8 ampC::Ptrp cI^+^*	pL promoter	[45]

**Table 2 ijms-23-03823-t002:** Plasmids used and their main features.

Plasmid	Origin	Copy Number	Reference
pET series	pBR322	15–20	[30,101]
pGEX series	pBR322	15–20	[98]
pCOLD I	ColE1	15–20	[53,98]
pH6EX3	ColE1	15–20	[102,103]
TAGZyme pQE-2	ColE1	15–20	[58]
pTrxFus	ColE1	15–20	[45]
pJOE2702	pBR322	15–20	[104]
pMW7	pBR322 derivative	High copy	[105]
pQE30	ColE1	15–20	[72]
pZE12luc	ColE1	15–20	[51]
pWaldo–GFPe	pET derivative	15–20	[106]
pRSET-A	pUC	High copy (500–700)	[30]
pHAT20	pUC	High copy (500–700)	[107]
PDS56/RBII	pBR322	15–20	[108]
pTMH-6FH	pT7-5 derivative, ColE1	15–20	[109]
pPOW-B2	pJLflO1 derivative, pBR322	15–20	[110]
pGTSD12	pGEM-3 derivative, pUC	500–700	[111]
pHG240/4	ColE1	15–20	[112]
pNZ8048	RepA, RepC	~50	[113]
pRUN	pET derivative	15–20	[114]

**Table 3 ijms-23-03823-t003:** Strategy adopted for membrane protein expression.

Protein/Alias	Plasmid	Tag	Strain	Strategy	Function	Reference
SLC1A5/ASCT2	pCOLD-I	N-Ter 6His	Rosetta-gami2	Low temperature/glucose	ND	[53]
SLC2A1/GLUT1	pGTSD12	None	SR425	P_L_ promoter, membrane insertion	M	[74]
SLC3A2/4F2hc	pGEX-4T1	N-Ter-GST	Rosetta(DE3)pLysS	28 °C	ND	[61]
SLC5A1/SGLT1	pTMH-6FH	FLAG	BL-21	0.3 mM IPTG, 16 °C, 5 h	P	[126]
SLC6A4/SERT	TAGZyme pQE-2	N-Ter 8His	BL21 CodonPlus (DE3) RP	Recovery from membrane	S	[58]
SLC6A19/B0AT1	pCOLD-I	N-Ter 6His	BL21 codonPlus RIL	Low temperature/low IPTG/glucose	ND	[53]
SLC7A5/LAT1	pH6EX3	N-Ter 6His	Rosetta(DE3)pLysS	0.4 mM IPTG, 28 °C, 4 h	P	[61]
SLC17A5/Sialin	pET-28a(+)	N-Ter and C-Ter β domain	C43(DE3)	Membrane insertion, 1 mM IPTG, 18 °C, 16 h	M	[66]
SLC17A9/VNUT	pET-28a(+)	N-Ter β domain	C43(DE3)	Membrane insertion, 1 mM IPTG, 18 °C, 16 h	M	[66]
SLC18A3/VAChT	pET-28a(+)	N- and C-ter YbeL	C43(DE3)	Membrane insertion, 1 mM IPTG, 18 °C, 16 h	M	[20]
SLC22A4/OCTN1	pH6EX3	N-Ter 6His	Rosetta(DE3)pLysS	0.4 mM IPTG 28 °C, 6 h	P	[62]
SLC22A4/OCTN1	pH6EX3	N-Ter 6His	Lemo21(DE3)	Codon optimization, 0.4 mM IPTG 28 °C, 6 h	P	[127]
SLC22A5/OCTN2	pET-21a(+)	C-Ter 6His	Rosetta(DE3)pLysS	R2K mutation	N	[46]
SLC22A5/OCTN2	pET-41a(+)	GST	Rosetta(DE3)pLysS	28 °C, 6 h	N	[46]
SLC25A7/UCP1	pET-26b(+)	PelB/6His/TEV	BL21 CodonPlus (DE3)-RIPL	Auto-induction method	S,P	[55]
SLC25A7/UCP1	pET-21a(+)	N-Ter 6His	BL21(DE3)	1 mM IPTG, 37 °C, 3 h	S,P	[56]
SLC25A8/UCP2	pET-21a(+)	N-Ter 6His	BL21(DE3)	1 mM IPTG, 37 °C, 3 h	S,P	[56]
SLC25A8/UCP2	pET-21a(+)	None	BL21(DE3)	1 mM IPTG, 30 °C, 6 h	P	[123]
SLC25A8/UCP2	pMW172	None	C41	2 h, 1 mM IPTG at 37 °C	ND	[65]
SLC25A9/UCP3	pET-21d(+)	N-Ter 6His	BL21(DE3)	1 mM IPTG, 37 °C, 3 h	S,P	[56]
SLC25A9/UCP3	pET-21a(+)	None	BL21(DE3)	1 mM IPTG, 30 °C, 6 h	P	[123]
SLC25A9/UCP3	pET-24a(+)	None	BL21(DE3)	1 mM IPTG, 37 °C, 2 h	P	[122]
SLC25A12/Aralar1	pNZ8048	N-Ter 8His	*L. lactis* NZ9000	Codon optimization	C,P	[13]
SLC25A13/Citrin	pNZ8048	N-Ter 8His	*L. lactis* NZ9000	Codon optimization	C,P	[13]
SLC25A14/UCP5	pET-21a(+)	N-Ter 6His	BL21(DE3)	1 mM IPTG, 37 °C, 3 h	S,P	[56]
SLC25A15/ORNT1	pET-21a(+)	C-Ter 6His	C0214	0.4 mM IPTG, 28 °C 4 h	P	[128]
SLC25A17/ PMP34	pET-21b	T7	Rosetta-gami B	30 °C inclusion bodies	P	[63]
SLC25A18/GC2 SLC25A22/GC1	pRUN	None	C0214(DE3)	0.4 mM IPTG, 37 °C 4.5 h	P	[69]
SLC25A20/CACT	pMW7	None	C0214(DE3)	0.4 mM IPTG, 37 °C 4 h	P	[67]
SLC25A21/ODC	pRUN	None	C0214(DE3)	0.4 mM IPTG, 37 °C 4.5 h	P	[68]
SLC25A24/APC1 SLC25A25/APC3	pQE30	N-Ter 6His	M15(pREP4)	Manufacturer’s instructions	P	[72]
SLC25A26/SAMC	pRUN	None	BL-21 CodonPlus(DE3)-RIL	0.4 mM IPTG, 37 °C 4.5 h	P	[54]
SLC25A27/UCP4	pET-21a(+)	N-Ter 6His	BL21CodonPlus (DE3)-RIPL	1 mM IPTG, 37 °C, 3 h	S,P	[56]
SLC25A29/ORNT3	pRUN	None	Rosetta-gami B(DE3)	37 °C	P	[129]
SLC29A1/ENT1	pHAT20	3xFLAG	BL21(DE3)	<25 °C		[107]
SLC30A8/ZnT8	pTrxFus	N-Ter Thioredoxin	GI724	Inclusion bodies	ND	[45]
SLC30A8/ZnT8	pTrxFus	N-Ter Thioredoxin	GI698	Intracellular soluble fraction	ND	[45]
SLC35C1/FUCT1	pJOE2702	ompA/FLAG	BW25113(DE3)	Codon optimization, membrane insertion	M	[71]
SLC35F3	pZE12luc	None	MW25113	Membrane insertion	M	[51]
SLC38A9	pH6EX3	N-Ter 6His	Lemo21(DE3)	Codon optimization, 39 °C, 2 h	P	[70]
SLC52A2/RFVT2	pH6EX3	N-Ter 6His	Rosetta(DE3)	Codon optimization	P	[59,60]
ABCB10	pET19b	N-Ter 6His	Rosetta2, BL21-CodonPlus (DE3)-RIPL	Codon optimization	L,S	[52]
MDR1	pPOW-B2	None	UT5600	Membrane insertion	M	[73]
hTRPV3	pWaldo–GFPe	GFP	BL21(DE3)-R3- pRARE 2	1 mM IPTG, TB medium, at 18 °C,	B	[64]
EAG1	pRSET-A	N-Ter 6His	BL21(DE3)	1 mM IPTG, 37 °C, 4 h	ND	[130]
MiRP1	pET-21b	C-Ter 6His	BL21(DE3) CodonPlus RP	1 mM IPTG, 37 °C, 4 h	S	[57]
ROMK1	pET-28a(+)	Several tags	BL21(DE3)-pLysS	Codon optimization, membrane insertion	L	[124]
CLIC4	pET-22b(+)	C-Ter 6His	BL21(DE3)	1 mM IPTG, 37 °C, 6 h	C	[131]
MICU1	pGEX-6p-1	N-Ter GST	BL21 DE3	0.5 mM IPTG, 16 °C, 20 h	S	[50]
MICU2-NΔ84-CΔ28	pET-28a(+)	N-Ter 6His	BL21 DE3	0.5 mM IPTG, 16 °C, 20	S	[50]
hVDAC1	pET-21a(+)	C-Ter 6His	BL21(DE3)	1 mM IPTG, 37 °C, 3 h	N,PP	[121,132]
hVDAC1 hVDAC2	PDS56/RBII	C-Ter 6His	M15(pREP4)	1 mM IPTG, 37 °C, 5–6 h	PP,S	[133]
hVDAC3	pET-21a(+)	C-Ter 6His	BL21(DE3)	1 mM IPTG, 37 °C, 3 h	PP	[44]

B: black lipid membranes; C: Crystallization; L: lipid nanodiscs; M: membrane-targeting; N: NMR; ND: not determined; P: proteoliposomes; PP: planar phospholipid membranes; S: spectroscopic analyses.

## Data Availability

Not applicable.

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
