# Peer review of "Strategies for Successful Over-Expression of Human Membrane Transport Systems Using Bacterial Hosts: Future Perspectives"

_ijms, 2022, doi:10.3390/ijms23073823_

Round 1

Reviewer 1 Report

The article entitled “Strategies for successful over-expression of human membrane transport systems using bacterial hosts, future perspectives”, by Galluccio et al, shows a great compilation of research related to the expression of eukaryotic transport systems in bacterial expression systems. I think it is certainly a great work, of great interest to the growing community of researchers interested in the expression and purification of eukaryotic transporters and channels, both for structural and functional assays. Therefore, the review by Galluccio et al, shows in an orderly fashion a large number of successful cases, allowing the non-expert researcher to find a whole series of cases that could serve as a guide for the expression of new eukaryotic targets in bacterial systems.

However, throughout the text, there are innumerable spelling mistakes (see line 154: "to produces"; line 164: "resumes" or line 155 "human membrane problems") and grammatical constructions that make the text difficult to read. It is, therefore, necessary to have the text revised by a specialized editor to improve its readability of the text.

On the other hand, a series of minor points are suggested that should be corrected to improve the quality of the text:

- The format of the paragraphs of the article should be homogenized. The line spacing is not the same in all paragraphs throughout the text, and in the case of the paragraph beginning on line 152, the indentation is not the same as in the rest of the paragraphs.

- The titles of the different sections also need to be homogenized. In some cases, the whole name of the proteins (or families) is followed by the acronym and in other cases only the acronym is present.

- Homogenise the format of the tables. In most cases, the text is centered in the middle of each cell of the table, but in some cases (some cells of Table 1 and the first column of Table 2) the text is not centered.

- Homogenise the nomenclature. For example, 6 His, 6xHis, his tag, etc... Nickel and Lactose should not be capitalized. In vitro should be italicized, as should the names of some genes that appear in the text (e.g. lines 287-299).

- Line 40: There is an extra space between "membranes" and the dot.

- Line 99: "wide topology of molecules" should be changed to "wide variety of molecules" or something similar. Topology referring to molecules in a structural biology review can be confusing.

- Line 104: homogenize the reference to SLCs as a family or superfamily. Both concepts are mixed and can be confusing.

- Line 113: "Very few high-resolution SLC transporters". I do not entirely agree with this statement. It may be a bit biased and therefore I think it would be more objective to show the number of resolved unique human SLC structures in relation to the total number of existing human SLC proteins, for example.

- Line 434: A parenthesis is missing after reference 155.

- Line 463: A dot is missing between "membrane" and reference 160.

- Line 468: "SARS-Cov-2" should be changed to "SARS-CoV-2".

- Line 510: A comma is missing between "SLC18A1-A3" and the parenthesis.

- Line 700: TPP should be defined.

- Lines 726-729: The nomenclature of the proteins does not seem to be homogeneous. While in some cases there are parts of the nomenclature in subscript in other cases there are not. Please homogenize if this is the case.

- Line 764: "crystallized by X-ray diffraction" should be changed to " crystallized and solved by X-ray diffraction".

Author Response

Reviewer 1.

The article entitled “Strategies for successful over-expression of human membrane transport systems using bacterial hosts, future perspectives”, by Galluccio et al, shows a great compilation of research related to the expression of eukaryotic transport systems in bacterial expression systems. I think it is certainly a great work, of great interest to the growing community of researchers interested in the expression and purification of eukaryotic transporters and channels, both for structural and functional assays. Therefore, the review by Galluccio et al, shows in an orderly fashion a large number of successful cases, allowing the non-expert researcher to find a whole series of cases that could serve as a guide for the expression of new eukaryotic targets in bacterial systems.

However, throughout the text, there are innumerable spelling mistakes (see line 154: "to produces"; line 164: "resumes" or line 155 "human membrane problems") and grammatical constructions that make the text difficult to read. It is, therefore, necessary to have the text revised by a specialized editor to improve its readability of the text.

Response. We thank the reviewer for her/his kind and valuable comments. We have corrected the indicated grammar errors and amended the whole text with the help of a native speaker.

On the other hand, a series of minor points are suggested that should be corrected to improve the quality of the text:

The suggested corrections, detailed below, have been performed and are tracked in yellow throughout the text.

 - The format of the paragraphs of the article should be homogenized. The line spacing is not the same in all paragraphs throughout the text, and in the case of the paragraph beginning on line 152, the indentation is not the same as in the rest of the paragraphs.

Response. Done

 - The titles of the different sections also need to be homogenized. In some cases, the whole name of the proteins (or families) is followed by the acronym and in other cases only the acronym is present.

Response. The titles of the different sections have been homogenized

 - Homogenise the format of the tables. In most cases, the text is centered in the middle of each cell of the table, but in some cases (some cells of Table 1 and the first column of Table 2) the text is not centered.

Response. Done.

 - Homogenise the nomenclature. For example, 6 His, 6xHis, his tag, etc... Nickel and Lactose should not be capitalized. In vitro should be italicized, as should the names of some genes that appear in the text (e.g. lines 287-299).

Response. Done.

- Line 40: There is an extra space between "membranes" and the dot.

Response. Done.

- Line 99: "wide topology of molecules" should be changed to "wide variety of molecules" or something similar. Topology referring to molecules in a structural biology review can be confusing.

Response. Thanks for the suggestion that indeed is right. The correction has been performed.

 - Line 104: homogenize the reference to SLCs as a family or superfamily. Both concepts are mixed and can be confusing.

Response. The reference to SLCs has been homogenized to superfamily

- Line 113: "Very few high-resolution SLC transporters". I do not entirely agree with this statement. It may be a bit biased and therefore I think it would be more objective to show the number of resolved unique human SLC structures in relation to the total number of existing human SLC proteins, for example.

Response. The suggestion is useful. We have introduced a precise information in the section 2.1 (SLC).

- Line 434: A parenthesis is missing after reference 155.

Response. Done

 - Line 463: A dot is missing between "membrane" and reference 160.

Response. Done

 - Line 468: "SARS-Cov-2" should be changed to "SARS-CoV-2".

Response. Done

 - Line 510: A comma is missing between "SLC18A1-A3" and the parenthesis.

Response. Done

- Line 700: TPP should be defined.

Response. Done

- Lines 726-729: The nomenclature of the proteins does not seem to be homogeneous. While in some cases there are parts of the nomenclature in subscript in other cases there are not. Please homogenize if this is the case.

Response. Done

- Line 764: "crystallized by X-ray diffraction" should be changed to " crystallized and solved by X-ray diffraction".

Response. Done

Reviewer 2 Report

Synopsis:
Engineering of recombinant proteins is essential for robust biochemical and biophysical analyses to determine the mechanistic insight of proteins at molecular or atomic level. One of first bottlenecks of recombinant proteins is to choose an ideal expression host for large-scale purification. Due to lack of many eukaryotic machinery that is necessary to synthesize eukaryotic proteins, bacteria are usually not the desired expression hosts for human membrane proteins. Over the years, however, people have continued searching for ways to optimize ideal conditions for eukaryotic protein expression by bacterial hosts. Galluccio et al in this review article to summarize the recent progress of how researchers have used E. coli bacteria to express human membrane proteins that involve in well-known membrane transport system, i.e., solute carriers, ATP-dependent transporters, and ion channels. The tables in this manuscript provide a good summary of E. coli-expressed human transporters, and Section 3 described published stories that allow the authors to make the case here. However, some major concerns or corrections require the attention of the authors, and when addressed properly, this reviewer believes this would be a handy reference in studying human and other eukaryotic membrane proteins by bacterial expression system.

Concerns and suggestions:
1. In Section 1, the authors briefly described the issues of why E. coli has not been an ideal system to express human membrane proteins and only focused primarily on how to overcome the inclusion body problem. To illustrate all the problems and to highlight how genetic manipulations of E. coli strains improve the success, a graphic summary would enhance the readability of this review and help the general audience to grasp the general and/or unique differences by using alternative bacterial strains.

2. The authors included several members of SLC and ion channels, but only two ABC transporters. In Section 2, the authors mentioned “ATP-dependent transporters”, but ABC transporters are not the only kind of ATP-dependent transporters. This reviewer wonders why the authors limited their review with ABC transporters. In addition, only ABCB10 and ABCB1 were mentioned, so it appears that E. coli may not be suitable for human ABC transporters, and there seems to be no follow-up of the cited ABCB1 work. In fact, wrong topological arrangement of the transmembrane domains has been shown in E. coli-expressed ABCB1 (PMID: 11989822). The authors need to address such issue. Otherwise, ABCB10 as the sole example is not representative, and this reviewer suggests removing ABC transporters completely from this review. The authors can make some discussion later in the manuscript.

3. Eukaryotic membrane proteins have been more successfully expressed using non-bacterial hosts, due to various reasons, albeit some purification yields are low. From protein engineering and biotech standpoint, consistent and successful production of functional and physiologically relevant human membrane proteins from bacteria would make a great difference in the membrane transporter community. In Section 3, when the authors detailed individual examples, they only highlighted what was done using bacteria. It is not clear about how different it is if we compared all these E. coli expressed proteins with yeast/insect/mammalian cell-expressions or native proteins? This manuscript may not be convincing enough in its current version.

4. Table 3 should include another column to show how the transporters were functionally and/or structurally verified.

Author Response

Reviewer 2

Synopsis:
Engineering of recombinant proteins is essential for robust biochemical and biophysical analyses to determine the mechanistic insight of proteins at molecular or atomic level. One of first bottlenecks of recombinant proteins is to choose an ideal expression host for large-scale purification. Due to lack of many eukaryotic machinery that is necessary to synthesize eukaryotic proteins, bacteria are usually not the desired expression hosts for human membrane proteins. Over the years, however, people have continued searching for ways to optimize ideal conditions for eukaryotic protein expression by bacterial hosts. Galluccio et al in this review article to summarize the recent progress of how researchers have used E. coli bacteria to express human membrane proteins that involve in well-known membrane transport system, i.e., solute carriers, ATP-dependent transporters, and ion channels. The tables in this manuscript provide a good summary of E. coli-expressed human transporters, and Section 3 described published stories that allow the authors to make the case here. However, some major concerns or corrections require the attention of the authors, and when addressed properly, this reviewer believes this would be a handy reference in studying human and other eukaryotic membrane proteins by bacterial expression system.

Concerns and suggestions:
1. In Section 1, the authors briefly described the issues of why E. coli has not been an ideal system to express human membrane proteins and only focused primarily on how to overcome the inclusion body problem. To illustrate all the problems and to highlight how genetic manipulations of E. coli strains improve the success, a graphic summary would enhance the readability of this review and help the general audience to grasp the general and/or unique differences by using alternative bacterial strains.

Response. We thank the reviewer for the suggestion. A Figure with a graphic summary is inserted in the revised manuscript (Fig. 1).

  1. The authors included several members of SLC and ion channels, but only two ABC transporters. In Section 2, the authors mentioned “ATP-dependent transporters”, but ABC transporters are not the only kind of ATP-dependent transporters. This reviewer wonders why the authors limited their review with ABC transporters. In addition, only ABCB10 and ABCB1 were mentioned, so it appears that E. coli may not be suitable for human ABC transporters, and there seems to be no follow-up of the cited ABCB1 work. In fact, wrong topological arrangement of the transmembrane domains has been shown in E. coli-expressed ABCB1 (PMID: 11989822). The authors need to address such issue. Otherwise, ABCB10 as the sole example is not representative, and this reviewer suggests removing ABC transporters completely from this review. The authors can make some discussion later in the manuscript.

Response. The issue suggested by the reviewer has been addressed in the MDR1/P-gp section.

  1. Eukaryotic membrane proteins have been more successfully expressed using non-bacterial hosts, due to various reasons, albeit some purification yields are low. From protein engineering and biotech standpoint, consistent and successful production of functional and physiologically relevant human membrane proteins from bacteria would make a great difference in the membrane transporter community. In Section 3, when the authors detailed individual examples, they only highlighted what was done using bacteria. It is not clear about how different it is if we compared all these E. coli expressed proteins with yeast/insect/mammalian cell-expressions or native proteins? This manuscript may not be convincing enough in its current version.

Response. The reviewer is right. This issue is only partially dealt with in the Introduction. Therefore, a sentence has been introduced in the revised manuscript.

  1. Table 3 should include another column to show how the transporters were functionally and/or structurally verified.

Response. An additional column has been added to the figure to show how function has been verified.

Reviewer 3 Report

Membrane transport systems are key element for cell life since they regulated the flux of nutrients, metabolites, and toxic compounds through cell and organelles membranes. This review summarizes the many strategies exploited for achieving the expression of human transport systems in Escherichya coli. It furnishes a very good guide for helping scientists studying eukaryotic transporter. E. coli can potentially lead to the production of milligrams expressed proteins at low costs for both structural and functional studies. Therefore, starting from the most used BL21 strain, different modifications have been introduced for achieving novel properties allowing good expression. Specific features of the modified strains are described in the review.

 Importantly, although the development of single particle cryoEM revolutionize structure determination of human transport proteins, the study of a transporter is far from its completion because structural dynamics, structure/function characterization and site-specific mutagenesis  required to understand the molecular mechanism and find the link of transporter defects with human pathology. Hence, I recommend publication

Minor comments:

  1. Genes names are traditionally writthen in Italics (for example lacY). Please correct all gene names in the review.
  2. Page 14, 401, … the 3D-structures of…
  3. Page 15, 444, …could then be over-expressed in Rosetta(…
  4. In-vivo and in-vitro should be in italics. Please correct through the review.

Author Response

Reviewer 3

Membrane transport systems are key element for cell life since they regulated the flux of nutrients, metabolites, and toxic compounds through cell and organelles membranes. This review summarizes the many strategies exploited for achieving the expression of human transport systems in Escherichya coli. It furnishes a very good guide for helping scientists studying eukaryotic transporter. E. coli can potentially lead to the production of milligrams expressed proteins at low costs for both structural and functional studies. Therefore, starting from the most used BL21 strain, different modifications have been introduced for achieving novel properties allowing good expression. Specific features of the modified strains are described in the review.

 Importantly, although the development of single particle cryoEM revolutionize structure determination of human transport proteins, the study of a transporter is far from its completion because structural dynamics, structure/function characterization and site-specific mutagenesis  required to understand the molecular mechanism and find the link of transporter defects with human pathology. Hence, I recommend publication

Minor comments:

  1. Genes names are traditionally writthen in Italics (for example lacY). Please correct all gene names in the review.
  2. Page 14, 401, … the 3D-structures of…
  3. Page 15, 444, …could then be over-expressed in Rosetta(…
  4. In-vivo and in-vitro should be in italics. Please correct through the review.

Response. We thank the reviewer for the nice comments and the suggestion. All the advised corrections have been performed.

Round 2

Reviewer 2 Report

The authors have addressed or adopted suggestions from the first review. As for ABC transporters, I still have doubts about using bacteria for recombinant proteins. ABCB10 works, likely due to its nature as a mitochondrial protein whose properties resemble those in bacteria. It remains to be seen whether new technology would make E. coli a more favorable host.

Author Response

The authors have addressed or adopted suggestions from the first review. As for ABC transporters, I still have doubts about using bacteria for recombinant proteins. ABCB10 works, likely due to its nature as a mitochondrial protein whose properties resemble those in bacteria. It remains to be seen whether new technology would make E. coli a more favorable host.

Response. We have introduced a new sentence in section 4.17 based on the reviewer suggetions.